# Uniform Error Bounds for Gaussian Process Regression with Application to Safe Control

**Armin Lederer**
Technical University of Munich
armin.lederer@tum.de

**Jonas Umlauft**
Technical University of Munich
jonas.umlauft@tum.de

**Sandra Hirche**
Technical University of Munich
hirche@tum.de

## Abstract

Data-driven models are subject to model errors due to limited and noisy training data. Key to the application of such models in safety-critical domains is the quantification of their model error. Gaussian processes provide such a measure and uniform error bounds have been derived, which allow safe control based on these models. However, existing error bounds require restrictive assumptions. In this paper, we employ the Gaussian process distribution and continuity arguments to derive a novel uniform error bound under weaker assumptions. Furthermore, we demonstrate how this distribution can be used to derive probabilistic Lipschitz constants and analyze the asymptotic behavior of our bound. Finally, we derive safety conditions for the control of unknown dynamical systems based on Gaussian process models and evaluate them in simulations of a robotic manipulator.

## 1 Introduction

The application of machine learning techniques in control tasks bears significant promises. The identification of highly nonlinear systems through supervised learning techniques [1] and the automated policy search in reinforcement learning [2] enables the control of complex unknown systems. Nevertheless, the application in safety-critical domains, like autonomous driving, robotics or aviation is rare. Even though the data-efficiency and performance of self-learning controllers is impressive, engineers still hesitate to rely on learning approaches if the physical integrity of systems is at risk, in particular, if humans are involved. Empirical evaluations, e.g. for autonomous driving [3], are available, however, this might not be sufficient to reach the desired level of reliability and autonomy.

Limited and noisy training data lead to imperfections in data-driven models [4]. This makes the quantification of the uncertainty in the model and the knowledge about a model's ignorance key for the utilization of learning approaches in safety-critical applications. Gaussian process models provide this measure for their own imprecision and therefore gained attention in the control community [5, 6, 7]. These approaches heavily rely on error bounds of Gaussian process regression and are therefore limited by the strict assumptions made in previous works on GP uniform error bounds [8, 9, 10, 11].

The main contribution of this paper is therefore the derivation of a novel GP uniform error bound, which requires less prior knowledge and assumptions than previous approaches and is therefore applicable to a wider range of problems. Furthermore, we derive a Lipschitz constant for the samples of GPs and investigate the asymptotic behavior in order to demonstrate that arbitrarily small error bounds can be guaranteed with sufficient computational resources and data. The proposed GP bounds are employed to derive safety guarantees for unknown dynamical systems which are controlled based

on a GP model. By employing Lyapunov theory [12], we prove that the closed-loop system - here we take a robotic manipulator as example - converges to a small fraction of the state space and can therefore be considered as safe.

The remainder of this paper is structured as follows: We briefly introduce Gaussian process regression and discuss related error bounds in Section 2. The novel proposed GP uniform error bound, the probabilistic Lipschitz constant and the asymptotic analysis are presented in Section 3. In Section 4 we show safety of a GP model based controller and evaluate it on a robotic manipulator in Section 5.

## 2 Background

### 2.1 Gaussian Process Regression and Uniform Error Bounds

Gaussian process regression is a Bayesian machine learning method based on the assumption that any finite collection of random variables[1] $y_i \in \mathbb{R}$ follows a joint Gaussian distribution with prior mean 0 and covariance kernel $k : \mathbb{R}^d \times \mathbb{R}^d \to \mathbb{R}_+$ [13]. Therefore, the variables $y_i$ are observations of a sample function $f : \mathbb{X} \subset \mathbb{R}^d \to \mathbb{R}$ of the GP distribution perturbed by zero mean Gaussian noise with variance $\sigma_n^2 \in \mathbb{R}_{+,0}$. By concatenating $N$ input data points $\boldsymbol{x}_i$ in a matrix $\boldsymbol{X}_N$ the elements of the GP kernel matrix $\boldsymbol{K}(\boldsymbol{X}_N, \boldsymbol{X}_N)$ are defined as $K_{ij} = k(\boldsymbol{x}_i, \boldsymbol{x}_j)$, $i, j = 1, \ldots, N$ and $\boldsymbol{k}(\boldsymbol{X}_N, \boldsymbol{x})$ denotes the kernel vector, which is defined accordingly. The probability distribution of the GP at a point $\boldsymbol{x}$ conditioned on the training data concatenated in $\boldsymbol{X}_N$ and $\boldsymbol{y}_N$ is then given as a normal distribution with mean $\nu_N(\boldsymbol{x}) = \boldsymbol{k}(\boldsymbol{x}, \boldsymbol{X}_N)(\boldsymbol{K}(\boldsymbol{X}_N, \boldsymbol{X}_N) + \sigma_n^2 \boldsymbol{I}_N)^{-1} \boldsymbol{y}_N$ and variance $\sigma_N^2(\boldsymbol{x}, \boldsymbol{x}') = k(\boldsymbol{x}, \boldsymbol{x}') - \boldsymbol{k}(\boldsymbol{x}, \boldsymbol{X}_N)(\boldsymbol{K}(\boldsymbol{X}_N, \boldsymbol{X}_N) + \sigma_n^2 \boldsymbol{I}_N)^{-1} \boldsymbol{k}(\boldsymbol{X}_N, \boldsymbol{x}')$.

A major reason for the popularity of GPs and related approaches in safety critical applications is the existence of uniform error bounds for the regression error, which is defined as follows.

**Definition 2.1.** *Gaussian process regression exhibits a uniformly bounded error on a compact set $\mathbb{X} \subset \mathbb{R}^d$ if there exists a function $\eta(\boldsymbol{x})$ such that*

$$|\nu_N(\boldsymbol{x}) - f(\boldsymbol{x})| \leq \eta(\boldsymbol{x}) \quad \forall \boldsymbol{x} \in \mathbb{X}. \tag{1}$$

*If this bound holds with probability of at least $1 - \delta$ for some $\delta \in (0, 1)$, it is called a probabilistic uniform error bound.*

### 2.2 Related Work

For many methods closely related to Gaussian process regression, uniform error bounds are very common. When dealing with noise-free data, i.e. in interpolation of multivariate functions, results from the field of scattered data approximation with radial basis functions can be applied [14]. In fact, many of the results from interpolation with radial basis functions can be directly applied to noise-free GP regression with stationary kernels. The classical result in [15] employs Fourier transform methods to derive an error bound for functions in the reproducing kernel Hilbert space (RKHS) attached to the interpolation kernel. By additionally exploiting properties of the RKHS a uniform error bound with increased convergence rate is derived in [16]. Typically, this form of bound crucially depends on the so called power function, which corresponds to the posterior standard deviation of Gaussian process regression under certain conditions [17]. In [18], a $\mathcal{L}_p$ error bound for data distributed on a sphere is developed, while the bound in [19] extends existing approaches to functions from Sobolev spaces. Bounds for anisotropic kernels and the derivatives of the interpolant are developed in [20]. A Sobolev type error bound for interpolation with Matérn kernels is derived in [21]. Moreover, it is shown that convergence of the interpolation error implies convergence of the GP posterior variance.

Regularized kernel regression is a method which extends many ideas from scattered data interpolation to noisy observations and it is highly related to Gaussian process regression as pointed out in [17]. In fact, the GP posterior mean function is identical to kernel ridge regression with squared cost

function [13]. Many error bounds such as [22] depend on the empirical $\mathcal{L}_2$ covering number and the norm of the unknown function in the RKHS attached to the regression kernel. In [23], the effective dimension of the feature space, in which regression is performed, is employed to derive a probabilistic uniform error bound. The effect of approximations of the kernel, e.g. with the Nyström method, on the regression error is analyzed in [24]. Tight error bounds using empirical $\mathcal{L}_2$ covering numbers are derived under mild assumptions in [25]. Finally, error bounds for general regularization are developed in [26], which depend on regularization and the RKHS norm of the function.

Using similar RKHS-based methods for Gaussian process regression, probabilistic uniform error bounds depending on the maximal information gain and the RKHS norm have been developed in [8]. These constants pose a high hurdle which has prevented the rigorous application of this work in control and typically heuristic constants without theoretical foundations are applied, see e.g. [27]. While regularized kernel regression allows a wide range of observation noise distributions, the bound in [8] only holds for bounded sub-Gaussian noise. Based on this work an improved bound is derived in [9] in order to analyze the regret of an upper confidence bound algorithm in multi-armed bandit problems. Although these bounds are frequently used in safe reinforcement learning and control, they suffer from several issues. On the one hand, they depend on constants which are very difficult to calculate. While this is no problem for theoretical analysis, it prohibits the integration of these bounds into algorithms and often estimates of the constants must be used. On the other hand, they suffer from the general problem of RKHS approaches: The space of functions, for which the bounds hold, becomes smaller the smoother the kernel is [19]. In fact, the RKHS attached to a covariance kernel is usually small compared to the support of the prior distribution of a Gaussian process [28].

The latter issue has been addressed by considering the support of the prior distribution of the Gaussian process as belief space. Based on bounds for the suprema of GPs [29] and existing error bounds for interpolation with radial basis functions, a probabilistic uniform error bound for Kriging (alternative term for GP regression for noise-free training data) is derived in [30]. However, the uniform error of Gaussian process regression with noisy observations has not been analyzed with the help of the prior GP distribution to the best of our knowledge.

## 3 Probabilistic Uniform Error Bound

While probabilistic uniform error bounds for the cases of noise-free observations and the restriction to subspaces of a RKHS are widely used, they often rely on constants which are hard to determine and are typically limited to unnecessarily small function spaces. The inherent probability distribution of GPs, which is the largest possible function space for regression with a certain GP, has not been exploited to derive uniform error bounds for Gaussian process regression with noisy observations. Under the weak assumption of Lipschitz continuity of the covariance kernel and the unknown function, a directly computable probabilistic uniform error bound is derived in Section 3.1. We demonstrate how Lipschitz constants for unknown functions directly follow from the assumed distribution over the function space in Section 3.2. Finally, we show that an arbitrarily small error bound can be reached with sufficiently many and well-distributed training data in Section 3.3.

### 3.1 Exploiting Lipschitz Continuity of the Unknown Function

In contrast to the RKHS based approaches in [8, 9], we make use of the inherent probability distribution over the function space defined by Gaussian processes. We achieve this through the following assumption.

**Assumption 3.1.** *The unknown function $f(\cdot)$ is a sample from a Gaussian process $\mathcal{GP}(0, k(\boldsymbol{x}, \boldsymbol{x}'))$ and observations $y = f(\boldsymbol{x}) + \epsilon$ are perturbed by zero mean i.i.d. Gaussian noise $\epsilon$ with variance $\sigma_n^2$.*

This assumption includes abundant information about the regression problem. The space of sample functions $\mathcal{F}$ is limited through the choice of the kernel $k(\cdot, \cdot)$ of the Gaussian process. Using Mercer's decomposition [31] $\phi_i(\boldsymbol{x})$, $i = 1, \ldots, \infty$ of the kernel $k(\cdot, \cdot)$, this space is defined through

$$\mathcal{F} = \left\{ f(\boldsymbol{x}) : \ \exists \lambda_i, i = 1, \ldots, \infty \text{ such that } f(\boldsymbol{x}) = \sum_{i=1}^{\infty} \lambda_i \phi_i(\boldsymbol{x}) \right\}, \tag{2}$$

which contains all functions that can be represented in terms of the kernel $k(\cdot, \cdot)$. By choosing a suitable class of covariance functions $k(\cdot, \cdot)$, this space can be designed in order to incorporate

prior knowledge of the unknown function $f(\cdot)$. For example, for covariance kernels $k(\cdot, \cdot)$ which are universal in the sense of [32], continuous functions can be learned with arbitrary precision. Moreover, for the squared exponential kernel, the space of sample functions corresponds to the space of continuous functions on $\mathbb{X}$, while its RKHS is limited to analytic functions [28]. Furthermore, Assumption 3.1 defines a prior GP distribution over the sample space $\mathcal{F}$ which is the basis for the calculation of the posterior probability. The prior distribution is typically shaped by the hyperparameters of the covariance kernel $k(\cdot, \cdot)$, e.g. slowly varying functions can be assigned a higher probability than functions with high derivatives. Finally, Assumption 3.1 allows Gaussian observation noise which is in contrast to the bounded noise required e.g. in [8, 9].

In addition to Assumption 3.1, we need Lipschitz continuity of the kernel $k(\cdot, \cdot)$ and the unknown function $f(\cdot)$. We define the Lipschitz constant of a differentiable covariance kernel $k(\cdot, \cdot)$ as

$$L_k := \max_{\boldsymbol{x}, \boldsymbol{x}' \in \mathbb{X}} \left\| \left[ \begin{array}{ccc} \frac{\partial k(\boldsymbol{x}, \boldsymbol{x}')}{\partial x_1} & \cdots & \frac{\partial k(\boldsymbol{x}, \boldsymbol{x}')}{\partial x_d} \end{array} \right]^T \right\|. \tag{3}$$

Since most of the practically used covariance kernels $k(\cdot, \cdot)$, such as squared exponential and Matérn kernels, are Lipschitz continuous [13], this is a weak restriction on covariance kernels. However, it allows us to prove continuity of the posterior mean function $\nu_N(\cdot)$ and the posterior standard deviation $\sigma_N(\cdot)$, which is exploited to derive a probabilistic uniform error bound in the following theorem. The proofs for all following theorems can be found in the supplementary material.

**Theorem 3.1.** *Consider a zero mean Gaussian process defined through the continuous covariance kernel $k(\cdot, \cdot)$ with Lipschitz constant $L_k$ on the compact set $\mathbb{X}$. Furthermore, consider a continuous unknown function $f : \mathbb{X} \to \mathbb{R}$ with Lipschitz constant $L_f$ and $N \in \mathbb{N}$ observations $y_i$ satisfying Assumption 3.1. Then, the posterior mean function $\nu_N(\cdot)$ and standard deviation $\sigma_N(\cdot)$ of a Gaussian process conditioned on the training data $\{(\boldsymbol{x}_i, y_i)\}_{i=1}^N$ are continuous with Lipschitz constant $L_{\nu_N}$ and modulus of continuity $\omega_{\sigma_N}(\cdot)$ on $\mathbb{X}$ such that*

$$L_{\nu_N} \leq L_k \sqrt{N} \left\| (\boldsymbol{K}(\boldsymbol{X}_N, \boldsymbol{X}_N) + \sigma_n^2 \boldsymbol{I}_N)^{-1} \boldsymbol{y}_N \right\| \tag{4}$$

$$\omega_{\sigma_N}(\tau) \leq \sqrt{2\tau L_k \left( 1 + N \| (\boldsymbol{K}(\boldsymbol{X}_N, \boldsymbol{X}_N) + \sigma_n^2 \boldsymbol{I}_N)^{-1} \| \max_{\boldsymbol{x}, \boldsymbol{x}' \in \mathbb{X}} k(\boldsymbol{x}, \boldsymbol{x}') \right)}. \tag{5}$$

*Moreover, pick $\delta \in (0, 1)$, $\tau \in \mathbb{R}_+$ and set*

$$\beta(\tau) = 2 \log \left( \frac{M(\tau, \mathbb{X})}{\delta} \right) \tag{6}$$

$$\gamma(\tau) = (L_{\nu_N} + L_f) \tau + \sqrt{\beta(\tau)} \omega_{\sigma_N}(\tau). \tag{7}$$

*Then, it holds that*

$$P \left( |f(\boldsymbol{x}) - \nu_N(\boldsymbol{x})| \leq \sqrt{\beta(\tau)} \sigma_N(\boldsymbol{x}) + \gamma(\tau), \ \forall \boldsymbol{x} \in \mathbb{X} \right) \geq 1 - \delta. \tag{8}$$

The parameter $\tau$ is in fact the grid constant of a grid used in the derivation of the theorem. The error on the grid can be bounded by exploiting properties of the Gaussian distribution [8] resulting in a dependency on the number of grid points. Eventually, this leads to the constant $\beta(\tau)$ defined in (6) since the covering number $M(\tau, \mathbb{X})$ is the minimum number of points in a grid over $\mathbb{X}$ with grid constant $\tau$. By employing the Lipschitz constant $L_{\nu_N}$ and the modulus of continuity $\omega_{\sigma_N}(\cdot)$, which are trivially obtained due Lipschitz continuity of the covariance kernel $k(\cdot, \cdot)$, as well as the Lipschitz constant $L_f$, the error bound is extended to the complete set $\mathbb{X}$, which results in (8).

Note, that most of the equations in Theorem 3.1 can be directly evaluated. Although our expression for $\beta(\tau)$ depends on the covering number of $\mathbb{X}$, which is in general difficult to calculate, upper bounds can be computed trivially. For example, for a hypercubic set $\mathbb{X} \subset \mathbb{R}^d$ the covering number can be bounded by

$$M(\tau, \mathbb{X}) \leq \left( 1 + \frac{r}{\tau} \right)^d, \tag{9}$$

where $r$ is the edge length of the hypercube. Furthermore, (4) and (5) depend only on the training data and kernel expressions, which can be calculated analytically in general. Therefore, (8) can

be computed for fixed $\tau$ and $\delta$ if an upper bound for the Lipschitz constant $L_f$ of the unknown function $f(\cdot)$ is known. Prior bounds on the Lipschitz constant $L_f$ are often available for control systems, e.g. based on simplified first order physical models. However, we demonstrate a method to obtain probabilistic Lipschitz constants from Assumption 3.1 in Section 3.2. Therefore, it is trivial to compute all expressions in Theorem 3.1 or upper bounds thereof, which emphasizes the high applicability of Theorem 3.1 in safe control of unknown systems.

Moreover, it should be noted that $\tau$ can be chosen arbitrarily small such that the effect of the constant $\gamma(\tau)$ can always be reduced to an amount which is negligible compared to $\sqrt{\beta(\tau)}\sigma_N(\boldsymbol{x})$. Even conservative approximations of the Lipschitz constants $L_{\nu_N}$ and $L_f$ and a loose modulus of continuity $\omega_{\sigma_N}(\tau)$ do not affect the error bound (8) much since (6) grows merely logarithmically with diminishing $\tau$. In fact, even the bounds (4) and (5), which grow in the order of $\mathcal{O}(N)$ and $\mathcal{O}(N^{\frac{1}{2}})$, respectively, as shown in the proof of Theorem 3.3 and thus are unbounded, can be compensated such that a vanishing uniform error bound can be proven under weak assumptions in Section 3.3.

## 3.2 Probabilistic Lipschitz Constants for Gaussian Processes

If little prior knowledge of the unknown function $f(\cdot)$ is given, it might not be possible to directly derive a Lipschitz constant $L_f$ on $\mathbb{X}$. However, we indirectly assume a certain distribution of the derivatives of $f(\cdot)$ with Assumption 3.1. Therefore, it is possible to derive a probabilistic Lipschitz constant $L_f$ from this assumption, which is described in the following theorem.

**Theorem 3.2.** *Consider a zero mean Gaussian process defined through the covariance kernel $k(\cdot, \cdot)$ with continuous partial derivatives up to the fourth order and partial derivative kernels*

$$k^{\partial i}(\boldsymbol{x}, \boldsymbol{x}') = \frac{\partial^2}{\partial x_i \partial x_i'} k(\boldsymbol{x}, \boldsymbol{x}') \quad \forall i = 1, \ldots, d. \tag{10}$$

*Let $L_k^{\partial i}$ denote the Lipschitz constants of the partial derivative kernels $k^{\partial i}(\cdot, \cdot)$ on the set $\mathbb{X}$ with maximal extension $r = \max_{\boldsymbol{x}, \boldsymbol{x}' \in \mathbb{X}} \|\boldsymbol{x} - \boldsymbol{x}'\|$. Then, a sample function $f(\cdot)$ of the Gaussian process is almost surely continuous on $\mathbb{X}$ and with probability of at least $1 - \delta_L$, it holds that*

$$L_f = \left\| \left[ \begin{array}{c} \sqrt{2 \log\left(\frac{2d}{\delta_L}\right)} \max_{\boldsymbol{x} \in \mathbb{X}} \sqrt{k^{\partial 1}(\boldsymbol{x}, \boldsymbol{x})} + 12\sqrt{6}d \max\left\{ \max_{\boldsymbol{x} \in \mathbb{X}} \sqrt{k^{\partial 1}(\boldsymbol{x}, \boldsymbol{x})}, \sqrt{r L_k^{\partial 1}} \right\} \\ \vdots \\ \sqrt{2 \log\left(\frac{2d}{\delta_L}\right)} \max_{\boldsymbol{x} \in \mathbb{X}} \sqrt{k^{\partial d}(\boldsymbol{x}, \boldsymbol{x})} + 12\sqrt{6}d \max\left\{ \max_{\boldsymbol{x} \in \mathbb{X}} \sqrt{k^{\partial d}(\boldsymbol{x}, \boldsymbol{x})}, \sqrt{r L_k^{\partial d}} \right\} \end{array} \right] \right\| \tag{11}$$

*is a Lipschitz constant of $f(\cdot)$ on $\mathbb{X}$.*

Note that a higher differentiability of the covariance kernel $k(\cdot, \cdot)$ is required compared to Theorem 3.1. The reason for this is that the proof of Theorem 3.2 exploits the fact that the partial derivative $k^{\partial i}(\cdot, \cdot)$ of a differentiable kernel is again a covariance function, which defines a derivative Gaussian process [33]. In order to obtain continuity of the samples of these derivative processes, the derivative kernels $k^{\partial i}(\cdot, \cdot)$ must be continuously differentiable [34]. Using the metric entropy criterion [34] and the Borell-TIS inequality [35], we exploit the continuity of sample functions and bound their maximum value, which directly translates into the probabilistic Lipschitz constant (11).

Note that all the values required in (11) can be directly computed. The maximum of the derivative kernels $k^{\partial i}(\cdot, \cdot)$ as well as their Lipschitz constants $L_k^{\partial i}$ can be calculated analytically for many kernels. Therefore, the Lipschitz constant obtained with Theorem 3.2 can be directly used in Theorem 3.1 through application of the union bound. Since the Lipschitz constant $L_f$ has only a logarithmic dependence on the probability $\delta_L$, small error probabilities for the Lipschitz constant can easily be achieved.

**Remark 3.1.** *The work in [36] derives also estimates for the Lipschitz constants. However, they only take the Lipschitz constant of the posterior mean function, which neglects the probabilistic nature of the GP and thereby underestimates the Lipschitz constants of samples of the GP.*

## 3.3 Analysis of Asymptotic Behavior

In safe reinforcement learning and control of unknown systems an important question regards the existence of lower bounds for the learning error because they limit the achievable control

performance. It is clear that the available data and constraints on the computational resources pose such lower bounds in practice. However, it is not clear under which conditions, e.g. requirements of computational power, an arbitrarily low uniform error can be guaranteed. The asymptotic analysis of the error bound, i.e. investigation of the bound (8) in the limit $N \to \infty$ can clarify this question. The following theorem is the result of this analysis.

**Theorem 3.3.** *Consider a zero mean Gaussian process defined through the continuous covariance kernel $k(\cdot, \cdot)$ with Lipschitz constant $L_k$ on the set $\mathbb{X}$. Furthermore, consider an infinite data stream of observations $(\boldsymbol{x}_i, y_i)$ of an unknown function $f : \mathbb{X} \to \mathbb{R}$ with Lipschitz constant $L_f$ and maximum absolute value $\bar{f} \in \mathbb{R}_+$ on $\mathbb{X}$ which satisfies Assumption 3.1. Let $\nu_N(\cdot)$ and $\sigma_N(\cdot)$ denote the mean and standard deviation of the Gaussian process conditioned on the first $N$ observations. If there exists a $\epsilon > 0$ such that the standard deviation satisfies $\sigma_N(\boldsymbol{x}) \in \mathcal{O}\left(\log(N)^{-\frac{1}{2}-\epsilon}\right), \forall \boldsymbol{x} \in \mathbb{X}$, then it holds for every $\delta \in (0, 1)$ that*

$$P\left(\sup_{\boldsymbol{x} \in \mathbb{X}} \|\nu_N(\boldsymbol{x}) - f(\boldsymbol{x})\| \in \mathcal{O}(\log(N)^{-\epsilon})\right) \geq 1 - \delta. \tag{12}$$

In addition to the conditions of Theorem 3.1 the absolute value of the unknown function is required to be bounded by a value $\bar{f}$. This is necessary to bound the Lipschitz constant $L_{\nu_N}$ of the posterior mean function $\nu_N(\cdot)$ in the limit of infinite training data. Even if no such constant is known, it can be derived from properties of the GP under weak conditions similarly to Theorem 3.2. Based on this restriction, it can be shown that the bound of the Lipschitz constant $L_{\nu_N}$ grows at most with rate $\mathcal{O}(N)$ using the triangle inequality and the fact that the squared norm of the observation noise $\|\boldsymbol{\epsilon}\|^2$ follows a $\chi_N^2$ distribution with probabilistically bounded maximum value [37]. Therefore, we pick $\tau(N) \in \mathcal{O}(N^{-2})$ such that $\gamma(\tau(N)) \in \mathcal{O}(N^{-1})$ and $\beta(\tau(N)) \in \mathcal{O}(\log(N))$ which implies (12).

The condition on the convergence rate of the posterior standard deviation $\sigma_N(\cdot)$ in Theorem 3.3 can be seen as a condition for the distribution of the training data, which depends on the structure of the covariance kernel. In [38, Corollary 3.2], the condition is formulated as follows: Let $\mathbb{B}_\rho(\boldsymbol{x})$ denote a set of training points around $\boldsymbol{x}$ with radius $\rho > 0$, then the posterior variance converges to zero if there exists a function $\rho(N)$ for which $\rho(N) \leq k(\boldsymbol{x}, \boldsymbol{x})/L_k \ \forall N$, $\lim_{N \to \infty} \rho(N) = 0$ and $\lim_{N \to \infty} \left|\mathbb{B}_{\rho(N)}(\boldsymbol{x})\right| = \infty$ holds. This is achieved, e.g. if a constant fraction of all samples lies on the point $\boldsymbol{x}$. In fact, it is straightforward to derive a similar condition for the uniform error bounds in [8, 9]. However, due to their dependence on the maximal information gain, the required decrease rates depend on the covariance kernel $k(\cdot, \cdot)$ and are typically higher. For example, the posterior standard deviation of a Gaussian process with a squared exponential kernel must satisfy $\sigma_N(\cdot) \in \mathcal{O}\left(\log(N)^{-\frac{d}{2}-2}\right)$ for [8] and $\sigma_N(\cdot) \in \mathcal{O}\left(\log(N)^{-\frac{d+1}{2}}\right)$ for [9].

# 4 Safety Guarantees for Control of Unknown Dynamical Systems

Safety guarantees for dynamical systems, in terms of upper bounds for the tracking error, are becoming more and more relevant as learning controllers are applied in safety-critical applications like autonomous driving or robots working in close proximity to humans [39, 40, 4]. We therefore show how the results in Theorem 3.1 can be applied to control safely unknown dynamical systems. In Section 4.1 we propose a tracking control law for systems which are learned with GPs. The stability of the resulting controller is analyzed in Section 4.2.

## 4.1 Tracking Control Design

Consider the nonlinear control affine dynamical system

$$\dot{x}_1 = x_2, \qquad \dot{x}_2 = f(\boldsymbol{x}) + u, \tag{13}$$

with state $\boldsymbol{x} = [x_1 \ x_2]^\mathsf{T} \in \mathbb{X} \subset \mathbb{R}^2$ and control input $u \in \mathbb{U} \subseteq \mathbb{R}$. While the structure of the dynamics (13) is known, the function $f(\cdot)$ is not. However, we assume that it is a sample from a GP with kernel $k(\cdot, \cdot)$. Systems of the form (13) cover a large range of applications including Lagrangian dynamics and many physical systems.

The task is to define a policy $\pi : \mathbb{X} \to \mathbb{U}$ for which the output $x_1$ tracks the desired trajectory $x_d(t)$ such that the tracking error $\boldsymbol{e} = [e_1\ e_2]^\mathsf{T} = \boldsymbol{x} - \boldsymbol{x}_d$ with $\boldsymbol{x}_d = [x_d\ \dot{x}_d]^\mathsf{T}$ vanishes over time, i.e. $\lim_{t \to \infty} \|\boldsymbol{e}\| = 0$. For notational simplicity, we introduce the filtered state $r = \lambda e_1 + e_2$, $\lambda \in \mathbb{R}_+$.

A well-known method for tracking of control affine systems is feedback linearization [12], which aims for a model-based compensation of the non-linearity $f(\cdot)$ using an estimate $\hat{f}(\cdot)$ and then applies linear control principles for the tracking. The feedback linearizing policy reads as

$$u = \pi(\boldsymbol{x}) = -\hat{f}(\boldsymbol{x}) + \nu, \tag{14}$$

where the linear control law $\nu$ is the PD-controller

$$\nu = \ddot{x}_d - k_c r - \lambda e_2, \tag{15}$$

with control gain $k_c \in \mathbb{R}_+$. This results in the dynamics of the filtered state

$$\dot{r} = f(\boldsymbol{x}) - \hat{f}(\boldsymbol{x}) - k_c r. \tag{16}$$

Assuming training data of the real system $y_i = f(\boldsymbol{x}_i) + \epsilon$, $i = 1, \ldots, N$, $\epsilon \sim \mathcal{N}(0, \sigma_n^2)$ are available, we utilize the posterior mean function $\nu_N(\cdot)$ for the model estimate $\hat{f}(\cdot)$. This implies, that observations of $\dot{x}_2$ are corrupted by noise, while $\boldsymbol{x}$ is measured free of noise. This is of course debatable, but in practice measuring the time derivative is usually realized with finite difference approximations, which injects significantly more noise than a direct measurement.

## 4.2 Stability Analysis

Due to safety constraints, e.g. for robots interacting with humans, it is usually necessary to verify that the model $\hat{f}(\cdot)$ is sufficiently precise and the parameters of the controller $k_c$, $\lambda$ are chosen properly. These safety certificates can be achieved if there exists an upper bound for the tracking error as defined in the following.

**Definition 4.1** (Ultimate Boundedness). *The trajectory $\boldsymbol{x}(t)$ of a dynamical system $\dot{\boldsymbol{x}} = \boldsymbol{f}(\boldsymbol{x}, \boldsymbol{u})$ is globally ultimately bounded, if there exist a positive constants $b \in \mathbb{R}_+$ such that for every $a \in \mathbb{R}_+$, there is a $T = T(a, b) \in \mathbb{R}_+$ such that*

$$\|\boldsymbol{x}(t_0)\| \le a \quad \Rightarrow \quad \|\boldsymbol{x}(t)\| \le b, \ \forall t \ge t_0 + T.$$

Since the solutions $\boldsymbol{x}(t)$ cannot be computed analytically, a stability analysis is necessary, which allows conclusions regarding the closed-loop behavior without running the policy on the real system [12].

**Theorem 4.1.** *Consider a control affine system (13), where $f(\cdot)$ admits a Lipschitz constant $L_f$ on $\mathbb{X} \subset \mathbb{R}^d$. Assume that $f(\cdot)$ and the observations $y_i$, $i = 1, \ldots, N$, satisfy the conditions of Assumption 3.1. Then, the feedback linearizing controller (14) with $\hat{f}(\cdot) = \nu_N(\cdot)$ guarantees with probability $1 - \delta$ that the tracking error $\boldsymbol{e}$ converges to*

$$\mathbb{B} = \left\{ \boldsymbol{x} \in \mathbb{X} \,\middle|\, \|\boldsymbol{e}\| \le \frac{\sqrt{\beta(\tau)}\sigma_N(\boldsymbol{x}) + \gamma(\tau)}{k_c\sqrt{\lambda^2 + 1}} \right\}, \tag{17}$$

*with $\beta(\tau)$ and $\gamma(\tau)$ defined in Theorem 3.1.*

Based on Lyapunov theory, it can be shown that the tracking error converges if the feedback term $|k_c r|$ dominates the model error $|f(\cdot) - \hat{f}(\cdot)|$. As Theorem 3.1 bounds the model error, the set for which holds $|k_c r| > \sqrt{\beta(\tau)}\sigma_N(\boldsymbol{x}) + \gamma(\tau)$ can be computed. It can directly be seen, that the ultimate bound can be made arbitrarily small, by increasing the gains $\lambda, k_c$ or with more training points to decrease $\sigma_N(\cdot)$. Computing the set $\mathbb{B}$ allows to check whether the controller (14) adheres to the safety requirements.

# 5 Numerical Evaluation

We evaluate our theoretical results in two simulations.[2] In Section 5.1, we investigate the effect of applying Theorem 3.2 to determine a probabilistic Lipschitz constant for an unknown synthetic

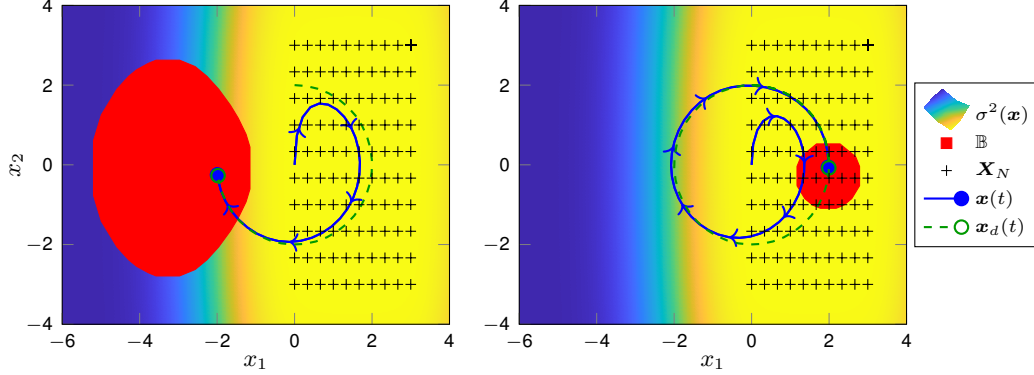

Figure 1: Snapshots of the state trajectory (blue) as it approaches the desired trajectory (green). In low uncertainty areas (yellow background), the set $\mathbb{B}$ (red) is significantly smaller then in high uncertainty areas (blue background).

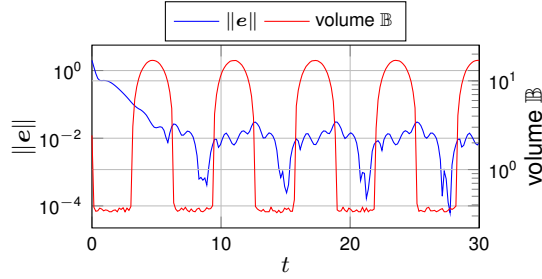

Figure 2: Higher uncertainty in the model leads to a larger ultimate bound (red). Similarly, the tracking error (blue) increases in areas with a less precise model.

system. Furthermore, we analyze the effect of unevenly distributed training samples on the tracking error bound from Theorem 4.1. In Section 5.2, we apply the feedback linearizing controller (14) to a tracking problem with a robotic manipulator.

## 5.1  Synthetic System with Unknown Lipschitz Constant $L_f$

As an example for a system of form (13), we consider $f(\boldsymbol{x}) = 1 - \sin(x_1) + \frac{1}{1+\exp(-x_2)}$. Based on a uniform grid over $[0\ 3] \times [-3\ 3]$ the training set is formed of 81 points with $\sigma_n^2 = 0.01$. The reference trajectory is a circle $x_d(t) = 2\sin(t)$ and the controller gains are $k_c = 2$ and $\lambda = 1$. We choose a probability of failure $\delta = 0.01$, $\delta_L = 0.01$ and set $\tau = 10^{-8}$. The state space is the rectangle $\mathbb{X} = [-6\ 4] \times [-4\ 4]$. A squared exponential kernel with automatic relevance determination is utilized, for which $L_k$ and $\max_{\boldsymbol{x}, \boldsymbol{x}' \in \mathbb{X}} k(\boldsymbol{x}, \boldsymbol{x}')$ is derived analytically for the optimized hyperparameters. We make use of Theorem 3.2 to estimate the Lipschitz constant $L_f$, and it turns out to be a conservative bound (factor $10 \sim 100$). However, this is not crucial, because $\tau$ can be chosen arbitrarily small and $\gamma(\tau)$ is dominated by $\sqrt{\beta(\tau)}\omega_{\sigma_N}(\tau)$. As Theorems 3.1 and 3.2 are subsequently utilized in this example, a union bound approximation can be applied to combine $\delta$ and $\delta_L$.

The results are shown in Figs. 1 and 2. Both plots show, that the safety bound here is rather conservative, which also results from the fact that the violation probability was set to $1\%$.

## 5.2  Robotic Manipulator with 2 Degrees of Freedom

We consider a planar robotic manipulator in the $z_1$-$z_2$-plane with 2 degrees of freedom (DoFs), with unit length and unit masses / inertia for all links. For this example, we consider $L_f$ to be known and extend Theorem 3.1 to the multidimensional case using the union bound. The state space is here four dimensional $[q_1\ \dot{q}_1\ q_2\ \dot{q}_2]$ and we consider $\mathbb{X} = [-\pi\ \pi]^4$. The 81 training points are distributed

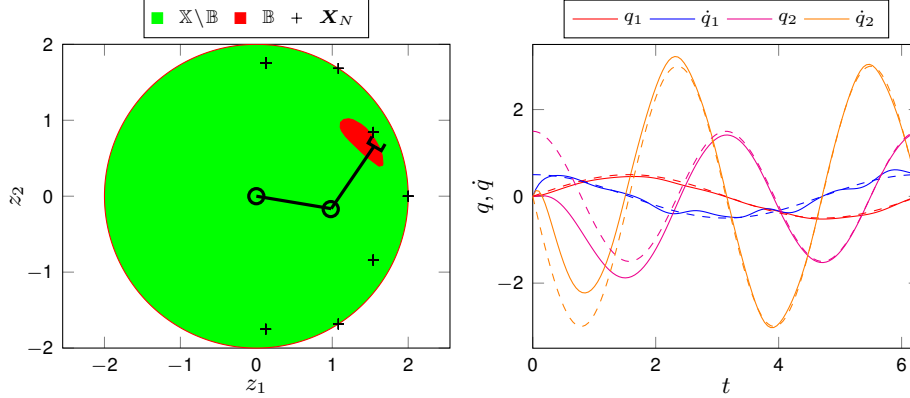

Figure 3: The task space of the robot (left) shows the robot is guaranteed to remain in $\mathbb{B}$ (red) after a transient phase. Hence, the remaining state space $\mathbb{X} \setminus \mathbb{B}$ (green) can be considered as safe. The joint angles and velocities (right) converge to the desired trajectories (dashed lines) over time.

in $[-1\ 1]^4$ and the control gain is $k_c = 7$, while other constants remain the same as in Section 5.1. The desired trajectories for both joints are again sinusoidal as shown in Fig. 3 on the right side. The robot dynamics are derived according to [41, Chapter 4].

Theorem 3.1 allows to derive an error bound in the joint space of the robot according to Theorem 4.1, which can be transformed into the task space as shown in Fig. 3 on the left. Thus, based on the learned (initially unknown) dynamics, it can be guaranteed, that the robot will not leave the depicted area and can thereby be considered as safe.

Previous error bounds for GPs are not applicable to this practical setting, because they i) do not allow the observation noise on the training data to be Gaussian [8], which is a common assumption in robotics, ii) utilize constants which cannot be computed efficiently (e.g. maximal information gain in [42]) or iii) make assumptions difficult to verify in practice (e.g. the RKHS norm of the unknown dynamical system [6]).

# 6 Conclusion

This paper presents a novel uniform error bound for Gaussian process regression. By exploiting the inherent probability distribution of Gaussian processes instead of the reproducing kernel Hilbert space attached to the covariance kernel, a wider class of functions can be considered. Furthermore, we demonstrate how probabilistic Lipschitz constants can be estimated from the GP distribution and derive sufficient conditions to reach arbitrarily small uniform error bounds. We employ the derived results to show safety bounds for a tracking control algorithm and evaluate them in simulation for a robotic manipulator.

### Acknowledgments

Armin Lederer gratefully acknowledges financial support from the German Academic Scholarship Foundation.

## Footnotes

[1]Notation: Lower/upper case bold symbols denote vectors/matrices and $\mathbb{R}_+/\mathbb{R}_{+,0}$ all real positive/non-negative numbers. $\mathbb{N}$ denotes all natural numbers, $\boldsymbol{I}_n$ the $n \times n$ identity matrix, the dot in $\dot{x}$ the derivative of $x$ with respect to time and $\|\cdot\|$ the Euclidean norm. A function $f(\boldsymbol{x})$ is said to admit a modulus of continuity $\omega : \mathbb{R}_+ \to \mathbb{R}_+$ if and only if $|f(\boldsymbol{x}) - f(\boldsymbol{x}')| \leq \omega(\|\boldsymbol{x} - \boldsymbol{x}'\|)$. The $\tau$-covering number $M(\tau, \mathbb{X})$ of a set $\mathbb{X}$ (with respect to the Euclidean metric) is defined as the minimum number of spherical balls with radius $\tau$ which is required to completely cover $\mathbb{X}$. Big $\mathcal{O}$ notation is used to describe the asymptotic behavior of functions.

[2]Matlab code is online available: `https://gitlab.lrz.de/ga68car/GPerrorbounds4safecontrol`

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
