[Supplementary Material · nips_2019_supplement.pdf]

# Supplementary Material

**Armin Lederer**
Technical University of Munich
`armin.lederer@tum.de`

**Jonas Umlauft**
Technical University of Munich
`jonas.umlauft@tum.de`

**Sandra Hirche**
Technical University of Munich
`hirche@tum.de`

## 1 Proof of Theorem 3.1

*Proof of Theorem 3.1.* We first prove the Lipschitz constant of the posterior mean $\nu_N(\boldsymbol{x})$ and the modulus of continuity of the standard deviation $\sigma_N(\boldsymbol{x})$, before we derive the bound of the regression error. The norm of the difference between the posterior mean $\nu_N(\boldsymbol{x})$ evaluated at two different points is given by

$$\|\nu_N(\boldsymbol{x}) - \nu_N(\boldsymbol{x}')\| = \|(\boldsymbol{k}(\boldsymbol{x}, \boldsymbol{X}_N) - \boldsymbol{k}(\boldsymbol{x}', \boldsymbol{X}_N))\, \boldsymbol{\alpha}\|$$

with

$$\boldsymbol{\alpha} = (\boldsymbol{K}(\boldsymbol{X}_N, \boldsymbol{X}_N) + \sigma_n^2 \boldsymbol{I}_N)^{-1} \boldsymbol{y}_N. \tag{1}$$

Due to the Cauchy-Schwarz inequality and the Lipschitz continuity of the kernel we obtain

$$\|\nu_N(\boldsymbol{x}) - \nu_N(\boldsymbol{x}')\| \leq L_k \sqrt{N}\, \|\boldsymbol{\alpha}\|\, \|\boldsymbol{x} - \boldsymbol{x}'\|,$$

which proves Lipschitz continuity of the mean $\nu_N(\boldsymbol{x})$. In order to calculate a modulus of continuity for the posterior standard deviation $\sigma_N(\boldsymbol{x})$ observe that the difference of the variance at two points $\boldsymbol{x}, \boldsymbol{x}' \in \mathbb{X}$ can be expressed as

$$|\sigma_N^2(\boldsymbol{x}) - \sigma_N^2(\boldsymbol{x}')| = |\sigma_N(\boldsymbol{x}) - \sigma_N(\boldsymbol{x}')||\sigma_N(\boldsymbol{x}) + \sigma_N(\boldsymbol{x}')|. \tag{2}$$

Since the standard deviation is positive semidefinite we have

$$|\sigma_N(\boldsymbol{x}) + \sigma_N(\boldsymbol{x}')| \geq |\sigma_N(\boldsymbol{x}) - \sigma_N(\boldsymbol{x}')| \tag{3}$$

and hence, we obtain

$$|\sigma_N^2(\boldsymbol{x}) - \sigma_N^2(\boldsymbol{x}')| \geq |\sigma_N(\boldsymbol{x}) - \sigma_N(\boldsymbol{x}')|^2. \tag{4}$$

Therefore, it is sufficient to bound the difference of the variance at two points $\boldsymbol{x}, \boldsymbol{x}' \in \mathbb{X}$ and take the square root of the resulting expression. Due to the Cauchy-Schwarz inequality and Lipschitz continuity of $k(\cdot, \cdot)$ the absolute value of the difference of the variance can be bounded by

$$|\sigma_N^2(\boldsymbol{x}) - \sigma_N^2(\boldsymbol{x}')| \leq 2L_k \|\boldsymbol{x} - \boldsymbol{x}'\|$$
$$+ \|\boldsymbol{k}(\boldsymbol{x}, \boldsymbol{X}_N) - \boldsymbol{k}(\boldsymbol{x}', \boldsymbol{X}_N)\| \, \|(\boldsymbol{K}(\boldsymbol{X}_N, \boldsymbol{X}_N) + \sigma_n^2 \boldsymbol{I}_N)^{-1}\| \, \|\boldsymbol{k}(\boldsymbol{X}_N, \boldsymbol{x}) + \boldsymbol{k}(\boldsymbol{X}_N, \boldsymbol{x}')\|. \tag{5}$$

On the one hand, we have

$$\|\boldsymbol{k}(\boldsymbol{x}, \boldsymbol{X}_N) - \boldsymbol{k}(\boldsymbol{x}', \boldsymbol{X}_N)\| \leq \sqrt{N} L_k \|\boldsymbol{x} - \boldsymbol{x}'\| \tag{6}$$

due to Lipschitz continuity of $k(\boldsymbol{x}, \boldsymbol{x}')$. On the other hand we have

$$\|\boldsymbol{k}(\boldsymbol{x}, \boldsymbol{X}_N) + \boldsymbol{k}(\boldsymbol{x}', \boldsymbol{X}_N)\| \leq 2\sqrt{N} \max_{\boldsymbol{x}, \boldsymbol{x}' \in \mathbb{X}} k(\boldsymbol{x}, \boldsymbol{x}'). \tag{7}$$

The modulus of continuity $\omega_{\sigma_N}(\tau)$ follows from substituting (6) and (7) in (5) and taking the square root of the resulting expression. Finally, we prove the probabilistic uniform error bound by exploiting the fact that for every grid $\mathbb{X}_\tau$ with $|\mathbb{X}_\tau|$ grid points and

$$\max_{\boldsymbol{x}\in\mathbb{X}} \min_{\boldsymbol{x}'\in\mathbb{X}_\tau} \|\boldsymbol{x}-\boldsymbol{x}'\| \le \tau \tag{8}$$

it holds with probability of at least $1-|\mathbb{X}_\tau|e^{-\beta(\tau)/2}$ that [1]

$$|f(\boldsymbol{x})-\nu_N(\boldsymbol{x})| \le \sqrt{\beta(\tau)}\sigma_N(\boldsymbol{x}) \quad \forall \boldsymbol{x}\in\mathbb{X}_\tau. \tag{9}$$

Choose $\beta(\tau) = 2\log\left(\frac{|\mathbb{X}_\tau|}{\delta}\right)$, then

$$|f(\boldsymbol{x})-\nu_N(\boldsymbol{x})| \le \sqrt{\beta(\tau)}\sigma_N(\boldsymbol{x}) \quad \forall \boldsymbol{x}\in\mathbb{X}_\tau \tag{10}$$

holds with probability of at least $1-\delta$. Due to continuity of $f(\boldsymbol{x})$, $\nu_N(\boldsymbol{x})$ and $\sigma_N(\boldsymbol{x})$ we obtain

$$\min_{\boldsymbol{x}'\in\mathbb{X}_\tau} |f(\boldsymbol{x})-f(\boldsymbol{x}')| \le \tau L_f \quad \forall \boldsymbol{x}\in\mathbb{X} \tag{11}$$

$$\min_{\boldsymbol{x}'\in\mathbb{X}_\tau} |\nu_N(\boldsymbol{x})-\nu_N(\boldsymbol{x}')| \le \tau L_{\nu_N} \quad \forall \boldsymbol{x}\in\mathbb{X} \tag{12}$$

$$\min_{\boldsymbol{x}'\in\mathbb{X}_\tau} |\sigma_N(\boldsymbol{x})-\sigma_N(\boldsymbol{x}')| \le \omega_{\sigma_N}(\tau) \quad \forall \boldsymbol{x}\in\mathbb{X}. \tag{13}$$

Moreover, the minimum number of grid points satisfying (8) is given by the covering number $M(\tau,\mathbb{X})$. Hence, we obtain

$$P\left(|f(\boldsymbol{x})-\nu_N(\boldsymbol{x})| \le \sqrt{\beta(\tau)}\sigma_N(\boldsymbol{x}) + \gamma(\tau), \ \forall \boldsymbol{x}\in\mathbb{X}\right) \ge 1-\delta, \tag{14}$$

where

$$\beta(\tau) = 2\log\left(\frac{M(\tau,\mathbb{X})}{\delta}\right) \tag{15}$$

$$\gamma(\tau) = (L_f + L_{\nu_N})\tau + \sqrt{\beta(\tau)}\omega_{\sigma_N}(\tau). \tag{16}$$

$\square$

## 2 Proof of Theorem 3.2

In order to proof Theorem 3.2, several auxiliary results are necessary, which are derived in the following. The first lemma concerns the expected supremum of a Gaussian process.

**Lemma 2.1.** *Consider a Gaussian process with a continuously differentiable covariance function $k(\cdot,\cdot)$ and let $L_k$ denote its Lipschitz constant on the set $\mathbb{X}$ with maximum extension $r = \max_{\boldsymbol{x},\boldsymbol{x}'\in\mathbb{X}} \|\boldsymbol{x}-\boldsymbol{x}'\|$. Then, the expected supremum of a sample function $f(\boldsymbol{x})$ of this Gaussian process satisfies*

$$E\left[\sup_{\boldsymbol{x}\in\mathbb{X}} f(\boldsymbol{x})\right] \le 12\sqrt{6d} \max\left\{\max_{\boldsymbol{x}\in\mathbb{X}} \sqrt{k(\boldsymbol{x},\boldsymbol{x})}, \sqrt{rL_k}\right\}. \tag{17}$$

*Proof.* We prove this lemma by making use of the metric entropy criterion for the sample continuity of some version of a Gaussian process [2]. This criterion allows to bound the expected supremum of a sample function $f(\boldsymbol{x})$ by

$$\mathrm{E}\left[\sup_{\boldsymbol{x}\in\mathbb{X}} f(\boldsymbol{x})\right] \le \int_0^{\max_{\boldsymbol{x}\in\mathbb{X}} \sqrt{k(\boldsymbol{x},\boldsymbol{x})}} \sqrt{\log(N(\varrho,\mathbb{X}))}\mathrm{d}\varrho, \tag{18}$$

where $N(\varrho,\mathbb{X})$ is the $\varrho$-packing number of $\mathbb{X}$ with respect to the covariance pseudo-metric

$$d_k(\boldsymbol{x},\boldsymbol{x}') = \sqrt{k(\boldsymbol{x},\boldsymbol{x}) + k(\boldsymbol{x}',\boldsymbol{x}') - 2k(\boldsymbol{x},\boldsymbol{x}')}. \tag{19}$$

Instead of bounding the $\varrho$-packing number, we bound the $\varrho/2$-covering number, which is known to be an upper bound. The covering number can be easily bounded by transforming the problem of

covering $\mathbb{X}$ with respect to the pseudo-metric $d_k(\cdot, \cdot)$ into a coverage problem in the original metric of $\mathbb{X}$. For this reason, define

$$\psi(\varrho') = \sup_{\substack{\boldsymbol{x}, \boldsymbol{x}' \in \mathbb{X} \\ \|\boldsymbol{x} - \boldsymbol{x}'\|_\infty \leq \varrho'}} d_k(\boldsymbol{x}, \boldsymbol{x}'), \tag{20}$$

which is continuous due to the continuity of the covariance kernel $k(\cdot, \cdot)$. Consider the inverse function

$$\psi^{-1}(\varrho) = \inf \left\{ \varrho' > 0 : \ \psi(\varrho') > \varrho \right\}. \tag{21}$$

Continuity of $\psi(\cdot)$ implies $\varrho = \psi(\psi^{-1}(\varrho))$. In particular, this means that we can guarantee $d_k(\boldsymbol{x}, \boldsymbol{x}') \leq \frac{\varrho}{2}$ if $\|\boldsymbol{x} - \boldsymbol{x}'\| \leq \psi^{-1}(\frac{\varrho}{2})$. Due to this relationship it is sufficient to construct an uniform grid with grid constant $2\psi^{-1}(\frac{\varrho}{2})$ in order to obtain a $\varrho/2$-covering net of $\mathbb{X}$. Furthermore, the cardinality of this grid is an upper bound for the $\varrho/2$-covering number, i.e.

$$M(\varrho/2, \mathbb{X}) \leq \left\lceil \frac{r}{2\psi^{-1}(\frac{\varrho}{2})} \right\rceil^d. \tag{22}$$

Therefore, it follows that

$$N(\varrho, \mathbb{X}) \leq \left\lceil \frac{r}{2\psi^{-1}(\frac{\varrho}{2})} \right\rceil^d. \tag{23}$$

Due to the Lipschitz continuity of the covariance function, we can bound $\psi(\cdot)$ by

$$\psi(\varrho') \leq \sqrt{2L_k \varrho'}. \tag{24}$$

Hence, the inverse function satisfies

$$\psi^{-1}\left(\frac{\varrho}{2}\right) \geq \left(\frac{\varrho}{2\sqrt{2L_k}}\right)^2 \tag{25}$$

and consequently

$$N(\varrho, \mathbb{X}) \leq \left(1 + \frac{4rL_k}{\varrho^2}\right)^d \tag{26}$$

holds, where the ceil operator is resolved through the addition of 1. Substituting this expression in the metric entropy bound (18) yields

$$E\left[\sup_{\boldsymbol{x} \in \mathbb{X}} f(\boldsymbol{x})\right] \leq 12\sqrt{d} \int_0^{\max_{\boldsymbol{x} \in \mathbb{X}} \sqrt{k(\boldsymbol{x}, \boldsymbol{x})}} \sqrt{\log\left(1 + \frac{4rL_k}{\varrho^2}\right)} \mathrm{d}\varrho. \tag{27}$$

As shown in [3] this integral can be bounded by

$$\int_0^{\max_{\boldsymbol{x} \in \mathbb{X}} \sqrt{k(\boldsymbol{x}, \boldsymbol{x})}} \sqrt{\log\left(1 + \frac{4rL_k}{\varrho^2}\right)} \mathrm{d}\varrho \leq \sqrt{6} \max\left\{ \max_{\boldsymbol{x} \in \mathbb{X}} \sqrt{k(\boldsymbol{x}, \boldsymbol{x})}, \sqrt{rL_k} \right\} \tag{28}$$

which proves the lemma. $\qquad\qquad\qquad\qquad\qquad\qquad\qquad\qquad\qquad\qquad\qquad\qquad\qquad\qquad$ $\square$

Based on the expected supremum of Gaussian process it is possible to derive a high probability bound for the supremum of a sample function.

**Lemma 2.2.** *Consider a Gaussian process with a continuously differentiable covariance function $k(\cdot, \cdot)$ and let $L_k$ denote its Lipschitz constant on the set $\mathbb{X}$ with maximum extension $r = \max_{\boldsymbol{x}, \boldsymbol{x}' \in \mathbb{X}} \|\boldsymbol{x} - \boldsymbol{x}'\|$. Then, with probability of at least $1 - \delta_L$ the supremum of a sample function $f(\boldsymbol{x})$ of this Gaussian process is bounded by*

$$\sup_{\boldsymbol{x} \in \mathbb{X}} f(\boldsymbol{x}) \leq \sqrt{2 \log\left(\frac{1}{\delta_L}\right)} \max_{\boldsymbol{x} \in \mathbb{X}} \sqrt{k(\boldsymbol{x}, \boldsymbol{x})} + 12\sqrt{6d} \max\left\{ \max_{\boldsymbol{x} \in \mathbb{X}} \sqrt{k(\boldsymbol{x}, \boldsymbol{x})}, \sqrt{rL_k} \right\}. \tag{29}$$

*Proof.* We prove this lemma by exploiting the wide theory of concentration inequalities to derive a bound for the supremum of the sample function $f(\boldsymbol{x})$. We apply the Borell-TIS inequality [4]

$$P\left(\sup_{\boldsymbol{x}\in\mathbb{X}} f(\boldsymbol{x}) - E\left[\sup_{\boldsymbol{x}\in\mathbb{X}} f(\boldsymbol{x})\right] \geq c\right) \leq \exp\left(-\frac{c^2}{2\max_{\boldsymbol{x}\in\mathbb{X}} k(\boldsymbol{x},\boldsymbol{x})}\right). \tag{30}$$

Due to Lemma 2.1 we have

$$E\left[\sup_{\boldsymbol{x}\in\mathbb{X}} f(\boldsymbol{x})\right] \leq 12\sqrt{6d}\max\left\{\max_{\boldsymbol{x}\in\mathbb{X}}\sqrt{k(\boldsymbol{x},\boldsymbol{x})}, \sqrt{rL_k}\right\}. \tag{31}$$

The lemma follows from substituting (31) in (30) and choosing $c = \sqrt{2\log\left(\frac{1}{\delta_L}\right)}\max_{\boldsymbol{x}\in\mathbb{X}}\sqrt{k(\boldsymbol{x},\boldsymbol{x})}$. $\square$

Finally, we exploit the fact that the derivative of a sample function is a sample function from another Gaussian process to prove the high probability Lipschitz constant in Theorem 3.2.

*Proof of Theorem 3.2.* Continuity of the sample function $f(\boldsymbol{x})$ follows directly from [5, Theorem 5]. Furthermore, this theorem guarantees that the derivative functions $\frac{\partial}{\partial x_i} f(\boldsymbol{x})$ are samples from derivative Gaussian processes with covariance functions

$$k_{\partial i}(\boldsymbol{x}, \boldsymbol{x}') = \frac{\partial^2}{\partial x_i \partial x_i'} k(\boldsymbol{x}, \boldsymbol{x}'). \tag{32}$$

Therefore, we can apply Lemma 2.2 to each of the derivative processes and obtain with probability of at least $1 - \frac{\delta_L}{d}$

$$-L_{f_{\partial i}} \leq \sup_{\boldsymbol{x}\in\mathbb{X}} \frac{\partial}{\partial x_i} f(\boldsymbol{x}) \leq L_{f_{\partial i}}, \tag{33}$$

where

$$L_{f_{\partial i}} = \sqrt{2\log\left(\frac{2d}{\delta_L}\right)}\max_{\boldsymbol{x}\in\mathbb{X}}\sqrt{k_{\partial i}(\boldsymbol{x},\boldsymbol{x})} + 12\sqrt{6d}\max\left\{\max_{\boldsymbol{x}\in\mathbb{X}}\sqrt{k_{\partial i}(\boldsymbol{x},\boldsymbol{x})}, \sqrt{rL_k^{\partial i}}\right\} \tag{34}$$

and $L_k^{\partial i}$ is the Lipschitz constant of derivative kernel $k_{\partial i}(\boldsymbol{x}, \boldsymbol{x}')$. Applying the union bound over all partial derivative processes $i = 1, \ldots, d$ finally yields the result. $\square$

## 3 Proof of Theorem 3.3

*Proof of Theorem 3.3.* Due to Theorem 3.1 with $\beta_N(\tau) = 2\log\left(\frac{M(\tau,\mathbb{X})\pi^2 N^2}{3\delta}\right)$ and the union bound over all $N > 0$ it follows that

$$\sup_{\boldsymbol{x}\in\mathbb{X}} |f(\boldsymbol{x}) - \nu_N(\boldsymbol{x})| \leq \sqrt{\beta_N(\tau)}\sigma_N(\boldsymbol{x}) + \gamma_N(\tau) \quad \forall N > 0 \tag{35}$$

with probability of at least $1 - \delta/2$. A trivial bound for the covering number can be obtained by considering a uniform grid over the cube containing $\mathbb{X}$. This approach leads to

$$M(\tau, \mathbb{X}) \leq \left(1 + \frac{r}{\tau}\right)^d, \tag{36}$$

where $r = \max_{\boldsymbol{x},\boldsymbol{x}'\in\mathbb{X}} \|\boldsymbol{x} - \boldsymbol{x}'\|$. Therefore, we have

$$\beta_N(\tau) \leq 2d\log\left(1 + \frac{r}{\tau}\right) + 4\log(\pi N) - 2\log(3\delta). \tag{37}$$

Furthermore, the Lipschitz constant $L_{\nu_N}$ is bounded by

$$L_{\nu_N} \leq L_k\sqrt{N}\left\|(\boldsymbol{K}(\boldsymbol{X}_N, \boldsymbol{X}_N) + \sigma_n^2 \boldsymbol{I}_N)^{-1}\boldsymbol{y}_N\right\| \tag{38}$$

due to Theorem 3.1. Since the Gram matrix $K(X_N, X_N)$ is positive semidefinite and $f(\cdot)$ is bounded by $\bar{f}$, we can bound $\left\| (K(X_N, X_N) + \sigma_n^2 I_N)^{-1} y_N \right\|$ by

$$\left\| (K(X_N, X_N) + \sigma_n^2 I_N)^{-1} y_N \right\| \leq \frac{\|y_N\|}{\rho_{\min}(K(X_N, X_N) + \sigma_n^2 I_N)}$$
$$\leq \frac{\sqrt{N}\bar{f} + \|\xi_N\|}{\sigma_n^2}, \tag{39}$$

where $\xi_N$ is a vector of $N$ i.i.d. zero mean Gaussian random variables with variance $\sigma_n^2$. Therefore, it follows that $\frac{\|\xi_N\|^2}{\sigma_n^2} \sim \chi_N^2$. Due to [6], with probability of at least $1 - \exp(-\eta_N)$ we have

$$\|\xi_N\|^2 \leq \left(2\sqrt{N\eta_N} + 2\eta_N + N\right)\sigma_n^2. \tag{40}$$

Setting $\eta_N = \log(\frac{\pi^2 N^2}{3\delta})$ and applying the union bounds over all $N > 0$ yields

$$\left\| (K(X_N, X_N) + \sigma_n^2 I_N)^{-1} y_N \right\| \leq \frac{\sqrt{N}\bar{f} + \sqrt{2\sqrt{N\eta_N} + 2\eta_N + N}\sigma_n}{\sigma_n^2} \quad \forall N > 0 \tag{41}$$

with probability of at least $1 - \delta/2$. Hence, the Lipschitz constant of the posterior mean function $\nu_N(\cdot)$ satisfies with probability of at least $1 - \delta/2$

$$L_{\nu_N} \leq L_k \frac{N\bar{f} + \sqrt{N(2\sqrt{N\eta_N} + 2\eta_N + N)}\sigma_n}{\sigma_n^2} \quad \forall N > 0. \tag{42}$$

Since $\eta_N$ grows logarithmically with the number of training samples $N$, it holds that $L_{\nu_N} \in \mathcal{O}(N)$ with probability of at least $1 - \delta/2$. The modulus of continuity $\omega_{\sigma_N}(\cdot)$ of the posterior standard deviation can be bounded by

$$\omega_{\sigma_N}(\tau) \leq \sqrt{2L_k\tau\left(\frac{N \max_{\tilde{x}, \tilde{x}' \in \mathbb{X}} k(\tilde{x}, \tilde{x}')}{\sigma_n^2} + 1\right)} \tag{43}$$

because $\|(K(X_N, X_N) + \sigma_n^2 I_N)^{-1}\| \leq \frac{1}{\sigma_n^2}$. Due to the union bound (35) holds with probability of at least $1 - \delta$ with

$$\gamma_N(\tau) \leq \sqrt{2L_k\tau\beta(\tau)\left(\frac{N \max_{\tilde{x}, \tilde{x}' \in \mathbb{X}} k(\tilde{x}, \tilde{x}')}{\sigma_n^2} + 1\right)} + L_f\tau + L_k\frac{N\bar{f} + \sqrt{N(2\sqrt{N\eta_N} + 2\eta_N + N)}}{\sigma_n^2}\tau. \tag{44}$$

This function must converge to 0 for $N \to \infty$ in order to guarantee a vanishing regression error. This is only ensured if $\tau(N)$ decreases faster than $\mathcal{O}((N\log(N))^{-1})$. Therefore, set $\tau(N) \in \mathcal{O}(N^{-2})$ in order to guarantee

$$\lim_{N \to \infty} \gamma_N(\tau_N) = 0. \tag{45}$$

However, this choice of $\tau(N)$ implies that $\beta_N(\tau(N)) \in \mathcal{O}(\log(N))$ due to (37). Since there exists an $\epsilon > 0$ such that $\sigma_N(x) \in \mathcal{O}\left(\log(N)^{-\frac{1}{2}-\epsilon}\right), \forall x \in \mathbb{X}$ by assumption, we have

$$\sqrt{\beta_N(\tau(N))}\sigma_N(x) \in \mathcal{O}(\log(N)^{-\epsilon}) \quad \forall x \in \mathbb{X}, \tag{46}$$

which concludes the proof. $\qquad\square$

## 4  Proof of Theorem 4.1

Lyapunov theory provides the following statement [7].

**Lemma 4.1.** *A dynamical system $\dot{x} = f(x, u)$ is globally ultimately bounded to a set $\mathbb{B} \subset \mathbb{X}$, containing the origin, if there exists a positive definite (so called Lyapunov) function, $V : \mathbb{X} \to \mathbb{R}_{+,0}$, for which $\dot{V}(x) < 0$, for all $x \in \mathbb{X} \setminus \mathbb{B}$.*

This allows to proof Theorem 4.1 as following.

*Proof of Theorem 4.1.* Consider the Lyapunov function $V(\boldsymbol{x}) = \frac{1}{2}r^2$

$$\dot{V}(\boldsymbol{x}) = \frac{\partial V}{\partial r}\dot{r} = r\left(f(\boldsymbol{x}) - \hat{f}(\boldsymbol{x}) - \mathfrak{k}_c r\right) \leq |r||f(\boldsymbol{x}) - \nu_N(\boldsymbol{x})| - \mathfrak{k}_c|r|^2 \leq 0 \quad \forall|r| > \frac{f(\boldsymbol{x}) - \nu_N(\boldsymbol{x})}{\mathfrak{k}_c}$$

Based on Theorem 3.1, the model error is bounded with high probability, which allows to conclude

$$P\left(\dot{V}(\boldsymbol{x}) < 0 \,\forall\boldsymbol{x} \in \mathbb{X} \setminus \mathbb{B}\right) \geq 1 - \delta.$$

The global ultimate boundedness of the closed-loop system, is thereby shown according to Lemma 4.1.

$\square$

## 5 Report on Computational Complexity of the Numerical Evaluation

Simulations are performed in MATLAB 2019a on a i5-6200U CPU with 2.3GHz and 8GB RAM. The simulation in Sec. 5.1 took 77s and used 1 MB of workspace memory. The simulation in Sec. 5.2 took 39s and used 134 MB of workspace memory. The code is available as supplementary material.