[Reviews · NeurIPS 2019]

Reviewer 1



This paper is a joy to read: clear, insightful and well-written. EDIT: I'm happy with how the authors addressed these concerns in the rebuttal. It is also not starved for space. Given that, perhaps it is a good idea to add a definition of what a uniform error bound is. Readers may have a wrong working definition of it. This is not too severe, as the theorems are rigorously stated and do not _use_ the definition of a uniform bound. However, they _are_ uniform bounds and it would be good to explain why. In the experiments, you observe the dynamical systems with some noise at every time step, and do regression from one step to the next. This is very sensible, since in a dynamical system, the measurements are usually taken with the same sensors at every step. However, note that doing this violates Assumption 3.1. This is because now you are also observing the _inputs_ of your regression with noise. You observe x_i + ε_i, so you do not know x_i. In practice in the experiments the noise is so small that it does not matter: the bounds can still be heuristically applied, and will be only a little bit incorrect. Regarding Section 3.3, why is it in the paper? Is it to show off the power of Theorem 3.1? It doesn't seem connected to the rest. As I mentioned earlier, it is unclear how to make sure the variance follows the decrease schedule O(log(N)^(-1/2-ε)). You state that it "depends on the structure of the covariance kernel". It would be good to add an example to this section, using something simple such as the RBF or linear kernel.

Reviewer 2



The paper provides bounds on the Lipschitz constants of sample functions from Gaussian processes. It also provides bounds on the prediction errors and on the Lipschitz constants of the conditional means and variances. Then, an application is safe control is provided. Originality: The bounds for Gaussian processes are relatively similar to the existing ones, from the references that are given in the paper. The proofs also appear to use standard arguments, for instance the classical bound on the mean value of the supremum of a Gaussian process and the Borel TIS inequality. The authors write that a novelty of their bounds is that they do not rely on unknown quantities. More precisely, the previous bounds hold for fixed functions assumed to belong to RKHSs while their bounds hold in probability w.r.t. the distribution of the Gaussian process. This type of analysis can also have its limits, because the distribution of the Gaussian process (which may need to be selected arbitrarily) can assign very small probabilities to functions that look like the unknown function. Hence, I would not say that the framework of the paper is better (or worse) than the one from previous references. It is just of different nature. The authors write that some references, for instance [7], require bounded observation errors and exclude Gaussian errors. I am not sure that this is true, because I think that these references require Gaussian observation errors. Quality: The proofs appear to be correct but relatively standard. Clarity: The paper is rather clear. Perhaps the authors could explain better the statement of Theorem 3.3. The speak of a process generating infinitely many functions with given Lipschitz constant. This can not be a Gaussian process, since the Lipschitz constant of a Gaussian process is usually not a bounded random variable. Significance: I think the paper provides a welcome contribution to the analysis of Gaussian process extrema, of the prediction error and of safe control. The results are nevertheless not very different from previously existing ones.

Reviewer 3



This article presents bounds on the absolute difference between a sample from a (possibly noisy) Gaussian process (GP) and the predictive mean. The methodology relies on deriving probabilistic Lipschitz constants for the GP, and does not consider the RKHS associated with the covariance kernel as is generally done. As a result, the bounds are easier to use in practice, as shown on two examples. The proposed Lispschitz bounds on GPs are an interesting result by themselves. But it should be related to and compared with previous derivations, see, e.g., González, J., Dai, Z., Hennig, P., & Lawrence, N. (2016, May). Batch Bayesian optimization via local penalization. In Artificial intelligence and statistics (pp. 648-657). Concerning the main theorem 3.3, could you precise how to achieve the condition on \sigma_N in practice. Simply adding more training points (P7L252) in an arbitrary way would not be sufficient, expecially with noise. Overall the paper is clear and well-written, with potential for both theoretical works and practical ones. Minor point: - Proof if Theorem 3.2 on continuity of samples: early results on this can be found, e.g., in Cramer, H., & Leadbetter, M. R. (1967). Stationary and Related Stochastic Processes-Sample Function Properties and their Applications. Typos: P3L83: on the one hand? P3L87: kernel is usual small P7L272: The simulations ## Added after rebuttal I would like to thank the authors for their response, which addresses most of my concerns. Accordingly I increased my score.

[Author Response · NeurIPS 2019]

**Concerns regarding Theorem 3.3**   Thank you for raising the interesting question on the conditions for asymptotic convergence to which an answer is provided in [1]. There it read as follows: Let $\mathbb{B}_\rho(\boldsymbol{x})$ denote a set of training points around $\boldsymbol{x}$ with radius $\rho > 0$, then the posterior variance converges to zero if there exists a function $\rho(N)$ for which $\rho(N) \leq k(\boldsymbol{x}, \boldsymbol{x})/L_k$ $\forall N$, $\lim_{N \to \infty} \rho(N) = 0$, $\lim_{N \to \infty} \big|\mathbb{B}_{\rho(N)}(\boldsymbol{x})\big| = \infty$ holds ($L_k$: Lipschitz constant of $k(\cdot, \cdot)$). This is achieved e.g. if a constant fraction of all samples lies on the point $\boldsymbol{x}$. We will add the reference [1] and a discussion on the conditions in the paper. In order to address the concerns of Reviewer #2, we will improve the clarity of Theorem 3.3 by reformulating lines 190-191 as follows: "Furthermore, consider an infinite data stream of observations $(\boldsymbol{x}_i, y_i)$ of an unknown function $f : \mathbb{X} \to \mathbb{R}$ with ...". Making Theorem 3.3 quantitative as suggested by Reviewer #2 follows directly from its proof and we will substitute (11) by $P(\sup_{\boldsymbol{x} \in \mathbb{X}} \|\nu_N(\boldsymbol{x}) - f(\boldsymbol{x})\| \in \mathcal{O}(\log(N)^{-\epsilon})) \geq 1 - \delta$.

**Boundedness of (3) and (4)**   As expected by Reviewer #2, (3) and (4) grow with the order of $N^{\frac{1}{2}}$ (see supplementary material (42), (43)). Although unbounded, they grow slow enough to allow the proof of Theorem 3.3 such that the main contribution of these bounds lies in the asymptotic analysis. However, in practical applications there are various ways to estimate tighter constants such as e.g. global optimization. We will add a brief discussion on this in the updated paper.

**Noise on input data**   Reviewer #1 pointed out, that Assumption 3.1. might be violated in the control example due to noise on input data . However, in the presented setup, there is no noise on the input because $f(\cdot)$ does not map from current state to next state, but from the state $\boldsymbol{x}$ to the time derivative of state $x_2$. Thus input data $\boldsymbol{x}$ and output data $\dot{x}_2$ are measured with two different sensors. Here we made the assumption, that observations of $\dot{x}_2$ are corrupted by noise, while $\boldsymbol{x}$ is measured noise free, which is of course debatable. But in practice, measuring the time derivative is usually realized with finite difference approximations, which injects significantly more noise than a direct measurement. Therefore, Assumption 3.1 is valid for our experimental setup. We will include the given reasoning in the updated paper.

**Relation to existing approaches**   We disagree with Reviewer #2 regarding the originality and significance of our contribution. Even though the bounds in Theorem 3.1 and [2, Theorem 6] look similar, their practical applicability is very different. Once the prior is fixed, all parameters for (7) can be easily computed such that a reliable error bound can be determined. In contrast, [2, Theorem 6] requires the information gain and a bound on the RKHS norm, which is assumed to be known (belonging to an RKHS does not suffice to compute the uniform error bound). In practice, we have observed that these parameters pose a high hurdle which has prevented the rigorous application of this theorem in control applications and typically heuristic constants without theoretical foundations are applied, see e.g. [3]. Therefore, even though both approaches have limits regarding their assumptions, Theorem 3.1 can be rigorously applied in practice, whereas this has been an issue with [2, Theorem 6]. We thank Reviewer #3 for pointing out the previous work [4] which derives a Lipschitz bound approximation for GPs. Although we think this work suggests a valuable estimator for the Lipschitz constant, it does not provide any theoretical guarantees. We will discuss this difference in the updated paper.

**Previous work require bounded observation noise**   Reviewer # 2 argues, that previous work, e.g. [2] are capable of dealing with unbounded noise. Even though [2] generally uses Gaussian noise, the (for this work) most relevant result in [2, Theorem 6] mentions the condition that "the noise variables $\epsilon_t$ are uniformly bounded by $\sigma$".

**Minor comments**   Thanks for pointing out various typos, we will fix all of them. As suggested by Reviewer #1, we will add a definition of a uniform error bound, extend the proof sketch for Theorem 3.2 and add sketches in the same style for Theorem 3.1, 3.3 and 4.1. Furthermore, reviewer #3 asked to consider a more complex control example: Generally, this is possible within this framework, where (12) becomes $\dot{x}_1 = x_2$, $\dot{x}_2 = x_3$, $\cdots \dot{x}_d = f(\boldsymbol{x}) + u$, with $\boldsymbol{x} = [x_1 \ x_2 \ \cdots \ \boldsymbol{x}_d]^\intercal$ using a definition $r = [\boldsymbol{\lambda}^\intercal \ 1]e$ where the coefficients in $\boldsymbol{\lambda} \in \mathbb{R}^{d-1}$ are Hurwitz. The robotic example can also directly be extended to arbitrary degrees of freedoms, however, for the sake of focus on the main results on the error bounds, we would keep the current control examples.

# References

[1] A. Lederer, J. Umlauft, and S. Hirche, "Posterior variance analysis of Gaussian processes with application to average learning curves," *arXiv preprint: arXiv:1906.01404*, 2019.

[2] N. Srinivas, A. Krause, S. M. Kakade, and M. W. Seeger, "Information-theoretic regret bounds for Gaussian process optimization in the bandit setting," *IEEE Transactions on Information Theory*, vol. 58, no. 5, pp. 3250–3265, 2012.

[3] F. Berkenkamp, M. Turchetta, A. P. Schoellig, and A. Krause, "Safe model-based reinforcement learning with stability guarantees," in *Advances in Neural Information Processing Systems*, 2017, pp. 908–918.

[4] J. González, Z. Dai, P. Hennig, and N. Lawrence, "Batch bayesian optimization via local penalization," in *Artificial intelligence and statistics*, May 2016, pp. 648–657.


[Meta-Review · NeurIPS 2019]

An interesting theoretical bound on the error of the posterior mean of GPR by bounding Lipschitz constants. Unlike previous work such bound is easier to evaluate and therefore it can be more practical, as it has been shown by the application to safe RL. Given the importance of GPs to sequential decision making under uncertainty the paper will be of interest to many practitioners. Please note that this submission caused a huge amount of discussion around conference policy issues regarding slicing contributions: "Note that slicing contributions too thinly may result in submissions being deemed dual submissions. Specifically, a case of slicing too thinly may correspond to two submissions by the same authors that are so similar that publishing one would render the other too incremental to be accepted.” The authors need to be aware of this when submitting to neurips or to any other future ML conferences that have similar policies.